# Identifying Key Factors in Adherence and Dropouts in Active Physiotherapy in Children with Acute Leukemia: A Systematic Review with Meta-Analysis and Meta-Regression

**DOI:** 10.3390/healthcare13212766

**Published:** 2025-10-30

**Authors:** Laura Ramírez-Pérez, Noelia Moreno-Morales

**Affiliations:** 1Department of Physiotherapy, University of Málaga, 29071 Málaga, Spain; lrp@uma.es; 2The Institute of Biomedical Research and Nanomedicine Platform in Málaga (IBIMA—BIONAND Platform), 29010 Málaga, Spain

**Keywords:** leukemia, exercise, rehabilitation adherence, dropouts, children, resilience

## Abstract

**Highlights:**

**What are the main findings?**
Exercise modality stands out as a potential moderator of adherence in children with acute leukemia.The heterogeneity of the studies prevents to extraction of consistent conclusions about moderators of dropouts in this population.

**What is the implication of the main finding?**
Physiotherapists should select strength or gaming-based protocols for treating this type of patient.The design of active physiotherapy interventions for children with acute leukemia should prioritize motivation to enhance the adherence and outcomes.

**Abstract:**

**Background/Objectives**: Adherence to active physiotherapy programs in children suffering from cancer is essential to enhance the improvement generated by the treatment. Therefore, the main aim of this review was to identify the factors influencing adherence and dropout rates in exercise programs applied to children with acute lymphoblastic leukemia. **Methods**: A systematic review with meta-analysis and meta-regression was conducted. The search was performed using PubMed, Scopus, Embase, SPORTDiscus, and PEDro databases. Eligible studies included randomized controlled trials focusing on determining adherence in active physiotherapy programs compared to standard care. A meta-synthesis was performed together with a random-effects meta-analysis. Furthermore, a proportion meta-analysis was developed, dividing by exercise modality, and a multivariate regression was performed to determine what factors were able to moderate the dropout rates. **Results**: Thirteen studies were selected, including 654 patients. Of them, 8 studies opt for multicomponent exercise, 3 used strength, and 2 selected virtual reality-based treatment. Overall, dropout rates were similar between groups. However, dropout proportions varied by intervention type, with minor attritions in strength (8.6%) and exergaming interventions (8.7%) compared to multicomponent exercise programs (18.4%). Meta-regression did not identify statistically significant moderators of dropouts. **Conclusions**: The heterogeneity of the studies in this target population meant that no factor could be identified as a moderator of dropouts, but exercise modality stands out as a potential moderator of adherence. Therefore, future studies should develop and test adherence-enhancing strategies to facilitate clinical implementation.

## 1. Introduction

Acute lymphoblastic leukemia is a malignant disorder of lymphoid progenitor cells that affects both children and adults, with the highest incidence occurring between 2 and 5 years of age. It is the most common pediatric cancer, accounting for 25–30% of childhood tumors [1,2], and survival rates in higher-income countries have now risen to over 85% thanks to advances in chemotherapy and supportive care [3,4]. Despite these improvements, treatment-related adverse effects—including reduced bone mineral density, sarcopenia, fatigue, and declines in motor and cardiorespiratory performance—remain significant challenges to the long-term health and quality of life of survivors [5,6,7].

Within this context, exercise-based interventions and active physiotherapy have emerged as effective strategies to counteract such adverse effects. Randomized controlled trials have reported improvements in muscle strength, functional capacity, fatigue reduction, and psychosocial outcomes among children with acute lymphoblastic leukemia participating in structured physical activity programs [8,9]. The most frequently implemented modalities are multicomponent exercise programs, understood as the combination of several types of exercise, such as aerobic training, flexibility routines, and strength exercises, together with balance and coordination, which have demonstrated positive effects on endurance, mobility, and overall quality of life in pediatric oncological patients [10,11]. Moreover, strength training alone is also widely used in this population. It focuses on increasing muscle force production through progressive and controlled loads using body weight or external elements such as resistance bands or dumbbells, and has been demonstrated to be effective in counteracting muscle atrophy, increasing muscular performance, and reducing cancer-related fatigue [12]. Additionally, virtual-reality-based interventions, involving immersive, computer-generated environments in which children can play games while performing functional activities and physical exercise, are increasingly being incorporated in pediatric oncological rehabilitation due to their benefits in the reduction in symptoms and the improvement of functional performance, together with the positive effects on the psychological well-being in this population [13,14].

Nevertheless, adherence and dropout rates remain critical barriers to the effectiveness and sustainability of these interventions. In pediatric oncology, adherence is shaped by multiple factors, including disease severity, treatment burden, cancer-related fatigue, psychological status, family support, and the degree of professional supervision [15,16]. Moreover, dropout has been linked to both modifiable factors—such as lack of interest, logistical barriers, and program design—and non-modifiable determinants, including comorbidities and educational level [17]. However, protocols incorporating active video games (exergaming) appear to be a promising solution for this issue due to their playful and motivating characteristics [18], addressing these challenges that underscore the critical impact of adherence and retention on the success of active physiotherapy interventions in children with acute lymphoblastic leukemia.

For these reasons, several authors have highlighted that it could be very relevant to modify the way in which the physiotherapist designs the intervention programs to address these challenges successfully [19]. Nonetheless, there is a lack of scientific evidence in terms of the factors that they need to narrow the focus on, so a comprehensive review of these factors is needed to solve this problem and enhance the design of active physiotherapy intervention programs since identifying which exercise modalities and contextual factors best promote sustained participation is essential for the clinical translation of these programs and for optimizing outcomes in this vulnerable population [20,21].

Therefore, the main aim of the present review was to identify the factors influencing adherence and dropout rates in exercise programs for children with acute lymphoblastic leukemia, since understanding adherence factors is central to fostering resilience and sustained engagement in rehabilitation among pediatric oncology patients.

## 2. Materials and Methods

### 2.1. Search Strategy

This study was conducted following the Preferred Reporting Items for Systematic Reviews and Meta-Analysis (PRISMA) statement [22], and the protocol of the review was registered in PROSPERO (registration ID: CRD420251148007).

The authors searched the scientific literature from inception to 11 August 2025 using the PubMed, Scopus, Embase, SPORTDiscus, and PEDro databases to carry out this study, following the recommendations for conducting biomedical systematic reviews [23]. The search window covered the period from the publication of the first relevant article on this topic up to August 2025, ensuring a comprehensive inclusion of all available evidence. The search string was designed by combining keywords or relevant terms from other similar studies as follows: (“leukemia” OR “leukaemia”) AND (“exercise” OR “strength” OR “aerobic” OR “physical activity” OR “active physiotherapy”) AND (“child” OR “children” OR “childhood” OR “pediatric”). These terms were screened in the title and abstract in the same search string.

The studies identified in the search were selected after three stages by two independent researchers (L.R.P. and N.M.M.). In the first stage, duplicate records were first identified using TERA (The Evidence Research Accelerator) software (https://tera-tools.com/), and any remaining duplicates were removed manually. After that, the titles and abstracts were screened. Finally, the articles were analyzed in full text to assess their suitability for inclusion in the systematic review. In addition, the snowballing technique was used to identify studies that could be potentially included in the review.

### 2.2. Eligibility Criteria

The studies that accomplished the following inclusion criteria were selected a priori:Randomized controlled trials.Focused on assessing whether the adherence to active physiotherapy programs is different from the standard physiotherapy interventions.Performed in children under 18 years old.

Those studies that met the inclusion criteria but presented one of the following exclusion criteria were excluded from the study:Studies that combine active physiotherapy with other treatments.Studies with more than two intervention groups.Pilot studies.

Multi-arm and pilot studies were excluded to maintain methodological homogeneity and to avoid potential small-sample bias.

### 2.3. Data Extraction

Data extraction was systematically done by the two independent reviewers to collect the following data: (1) population profile, (2) sample size, (3) age, (4) sex, (5) intervention characteristics, (6) follow-up timeframe, (7) dropouts during the trial, and (8) adherence parameters if they are reported.

### 2.4. Study Quality Assessment

The internal validity of the included studies was assessed using the Physiotherapy Evidence Database scale (PEDro) [24] that includes eleven items to assess eligibility criteria, random and concealed allocation, baseline comparability, triple blinding (subjects, therapists, and evaluators), adequate follow-up, intention-to-treat analysis, between-group comparisons, point estimates, and measures of variability.

Furthermore, the risk of bias of the studies was analysed using the Cochrane tool of risk of bias for randomized controlled trials (RoB 2) [25], which evaluates five domains including the randomization process, the deviation from the intended intervention, the loss of data, the measurement of the outcome variables, and the selection of the reported results.

Two reviewers (L.R.-P. and N.M.-M.) independently assessed the internal validity and the risk of bias; discrepancies were resolved by discussion.

### 2.5. Data Analysis

A meta-analysis was conducted with random effects using Jamovi version 2.4.11, through the use of the MAJOR R 1.4.5. package using the total number of patients in each group, together with the dropouts for each group, dividing between the control group and exercise group. Additionally, a proportion meta-analysis was conducted to determine the influence of exercise type on dropouts, using only the intervention group. Moreover, the chi-square test was used to determine the differences in adherence between types of intervention. After that, a multivariate meta-regression was performed using R Studio (version 4.5.1.). The reasons for dropouts were included as potential moderators across studies. Death, non-compliant, lack of interest, concomitant diseases, schedule, family relocation, adverse events, and other factors were included as moderators.

The significance level established was *p* < 0.05, and heterogeneity was assessed using the I^2^ statistic. The data were represented using forest plots.

## 3. Results

### 3.1. Characteristics of Included Studies

Based on the eligibility criteria, 2651 studies were found, of which 1692 were analysed based on the title and abstract, and 306 were assessed in full text. After that, 9 randomized controlled trials were selected for inclusion, and 4 additional reports were included based on citation searching, resulting in a total of 13 included studies (Figure 1). Then, the PEDro scale was used to analyze the internal validity of those studies, which ranged from five to eight points (Table 1). In addition, the risk of bias was assessed using the RoB 2 tool, with twelve studies varying from low and moderate risk of bias (Figure 2).

Table 2 displays the characteristics of the studies, with a total sample of 654 patients ranging from 1 to 18 years. The authors included patients newly diagnosed [5,8,16,26,27,28], children receiving chemotherapy [29,30,31,32], and survivors [33,34,35]. Concerning the intervention, eight studies opt for a multicomponent exercise program for the experimental group, including strengthening, stretching, and aerobic exercises [5,8,26,27,29,31,32,35], while three studies decided to use a protocol more focused on strengthening [16,33,34], and the rest two studies used a virtual reality-based program to treat children by the use of several Nintendo Wii exergames [28,30].

**Table 1 healthcare-13-02766-t001:** PEDro internal validity.

	Eligibility Criteria	Random Allocation	Concealed Allocation	Baseline Comparability	Blind Subjects	Blind Therapists	Blind Assessors	Adequate Follow-Up	Intention-to-Treat Analysis	Between-Group Comparisons	Point Estimates and Variability	Total
Cox et al. (2018) [26]	1	1	1	0	0	0	1	1	0	1	1	6/10
Elnaggar et al. (2025) [33]	1	1	1	1	0	0	1	1	1	1	1	8/10
Elnaggar and Mohamed (2021) [34]	1	0	0	1	0	0	1	1	0	1	1	5/10
Gaser et al. (2022) [16]	1	1	0	1	0	0	0	1	1	1	1	6/10
Hartman et al. (2009) [5]	1	1	1	1	0	0	1	1	1	1	1	8/10
Marchese et al. (2004) [29]	1	1	1	1	0	0	1	1	0	1	1	7/10
Masoud et al. (2023) [30]	1	1	0	1	0	0	0	1	1	1	1	6/10
Moyer-Mileur et al. (2009) [8]	1	1	0	1	0	0	0	1	0	1	1	5/10
Saultier et al. (2021) [27]	1	1	0	1	0	0	0	1	0	1	1	5/10
Schmidt-Andersen et al. (2025) [31]	1	1	0	1	0	0	0	1	0	1	1	5/10
Tanir and Kuguoglu (2012) [35]	1	1	0	1	0	0	0	1	0	1	1	5/10
Tanriverdi et al. (2022) [28]	1	1	0	1	0	0	0	1	0	1	1	5/10
Waked and Albenasy (2018) [32]	1	1	1	1	0	0	0	1	0	1	1	6/10

**Table 2 healthcare-13-02766-t002:** Characteristics of the included studies.

Reference	Population Profile	Sample Size	Age	Sex (%)	Intervention		Time of Follow-Up	Adherence
					Control Group (CG)	Experimental Group (EG)		
Cox et al. (2018) [26]	Children newly diagnosed with acute lymphoblastic leukemia	N = 107 CG = 54 EG = 53	4–18 years	Male: 70 (65.4%) Female:37 (34.6%)	Passive ankle once weekly and trunk stretching at home five times weekly	Strength, range of motion, gross motor skills, and endurance once weekly	2.5 years	Data not specified
Elnaggar et al. (2025) [33]	Survivors of childhood acute lymphoblastic leukemia	N = 62 CG = 31 EG = 31	12–18 years CG: 14.35 ± 1.76 EG: 15.13 ± 1.94	Male: 34 (54.84%) Female: 28 (45.16%)	8 weeks of stretching, strengthening, and moderate aerobic exercises (50–70% intensity) three times weekly for 45 min	8 weeks of adaptive variable-resistance training with an isokinetic dynamometer in maximum voluntary knee flexor and extensor concentric actions, with three weekly sessions	8 weeks	CG: 91.67% EG: 95.83%
Elnaggar and Mohamed (2021) [34]	Survivors of acute lymphoblastic leukemia	N = 30 CG = 15 EG = 15	8–18 years CG: 12.87 ± 2.56 EG: 13.33 ± 3.13	Male: 19 (63.3%) Female: 11 (36.7%)	12-week standard care, including stretching, strengthening, and aerobic exercises (75% intensity) three times weekly, with sessions of 45 min	12-week plyometric exercise program with 3 weekly sessions of 45 min, including 10 lower-body aqua plyometric exercises	12 weeks	CG: 91.67% EG: 95.83%
Gaser et al. (2022) [16]	Acute lymphoblastic leukemia, myeloid leukemia, or non-Hodgkin lymphoma	N = 41 CG = 20 EG = 21	4–18 years CG: 9.7 ± 3.9 EG: 10.2 ± 4.2	Male: 27 (65.85%) Female: 14 (34.15%)	Standard care exercise program with 2–3 weekly sessions	Specific strength training combined with a standard care exercise program, with 2–3 weekly sessions	7 months	CG: 68% EG: 65%
Hartman et al. (2009) [5]	Diagnosis of acute lymphoblastic leukemia	N = 51 CG = 26 EG = 25	1–18 years CG: 6.2 EG: 5.3	Male: 30 (58.82%) Female: 21 (41.18%)	General advice on doing exercise	Specific home exercises including stretching, motor skills, and short-burst high-intensity exercises + a session every 6 weeks	2 years	Data not specified
Marchese et al. (2004) [29]	Children with acute lymphoblastic leukemia receiving chemotherapy	N = 28 CG = 15 EG = 13	4–18 years CG: 8.3 EG: 7.5	Male: 20 (80%) Female: 8 (20%)	No instructions related to physical fitness	12-week protocol with 5 sessions of physical therapy plus an exercise program at home including stretching 5 times per week, strengthening 3 times per week with functional exercises, and aerobic fitness daily.	12 weeks	Data no specified
Masoud et al. (2023) [30]	Children with acute lymphoblastic leukemia receiving chemotherapy	N = 46 CG = 23 EG = 23	6–14 years CG: 9.04 ± 2.29 EG: 9.00 ± 2.35	Male: 25 (55.56%) Female: 20 (44.44%)	Instructional session about the benefits of physical activity	3-week protocol using 23 Wii games: sports resorts, fit plus exergames, aerobic game, and balance games that require full-body engagement with an intensity of 50–70% with 2 weekly sessions of 60 min	5 weeks	CG: data not specified EG: 100%
Moyer-Mileur et al. (2009) [8]	Standard-risk acute lymphoblastic leukemia	N = 14 CG = 7 EG = 7	4–10 years CG: 5.9 ± 0.7 EG: 7.2 ± 0.7	Male: 7 (50%) Female: 7 (50%)	General advice for physical activity	12-month home-based exercise including strength, flexibility, endurance, recreational sports, and lifestyle activities, plus one monthly phone call and diet prescription	12 months	Data not specified
Saultier et al. (2021) [27]	Children and adolescents diagnosed with cancer (39% leukemia, 61% solid tumors)	N = 80 CG = 39 EG = 41	5–19 years CG: 11.2 ± 0.6 EG: 11.4 ± 0.6	Male: 46 (%) Female: 34 (%)	Recreational activities, which included: Board games, Storytelling Manual and creative activities Film evenings	A 6-month Physical Activity Program (PAP) including 30 exercise sessions (30–90 min) and 15 multi-activity sessions (90–240 min), such as dance, basketball, swimming, yoga, and outdoor camps.	12 months	Data not specified
Schmidt-Andersen et al. (2025) [31]	Children and adolescents diagnosed with cancer and receiving chemotherapy and/or irradiation during the first 6 months of treatment	N = 84 CG = 40 EG = 44	11.6 ± 3.7 years CG = 11.6 ± 3.4 years EG = 11.6 ± 3.9 years	Male: 51 (64%) Female: 33 (36%)	24-week Integrative Neuromuscular Training (INT), including supervised hospital-based and unsupervised home-based sessions.	24-week unsupervised home-based exercise program with a training log and suggested activities.	6 months	CG = 3% EG = 67%
Tanir and Kuguoglu (2012) [35]	Children with a diagnosis of acute lymphoblastic leukemia at least 1 year before the study	N = 41 CG = 21 EG = 20	8–12 years	Male: 24 (58.54%) Female: 17 (41.46%)	Standard care without exercise prescription	A 3-month program of home exercises, including mobility (5 days a week), strengthening (3 times a week), and aerobic (3 times a week) exercises, plus 3 supervised sessions	3 months	Data not specified
Tanriverdi et al. (2022) [28]	Children with acute lymphoblastic leukemia	N = 24 CG = 13 EG = 11	6–18 years	Male: 9 (37.50%) Female: 15 (62.5%)	Supportive care	Virtual reality-based aerobic exercises using Nintendo Wii Fit Plus were applied for 12 weeks, 2 times weekly with sessions of 45 min	12 weeks	Data not specified
Waked and Albenasy (2018) [32]	Children with acute lymphoblastic leukemia in the maintenance phase of chemotherapy	N = 46 CG = 23 EG = 23	6–14 years CG: 9.91 ± 2.09 EG: 9.26 ± 2.39	Male: 33 (71.7%) Female: 13 (28.3%)	Advised to stay as active as possible, no equipment or formal instructions	Supervised mixed program (aerobic + resistance with bands + flexibility), 2 sessions/week for 6 months, then 1 session/week for another 6 months; light–moderate intensity (RPE 3–6/10), 30–45 min per session; warm-up 5 min; progression based on tolerance	12 months	Data not specified

**Figure 2 healthcare-13-02766-f002:**
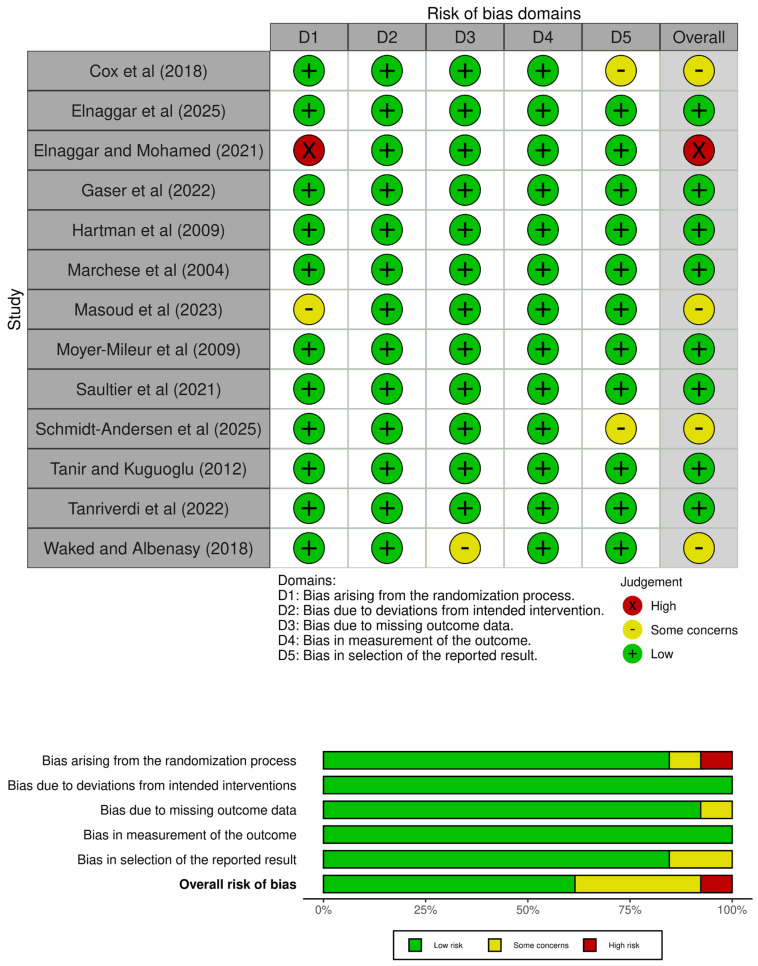
Risk of bias assessment [5,8,16,26,27,28,29,30,31,32,33,34,35].

### 3.2. Adherence

Only five of the thirteen studies reported data about participant adherence to the intervention. All of them defined adherence as the percentage of sessions attended and completed of the total number of planned sessions. Considering the heterogeneity in the studies that reported adherence, a meta-analysis was not conducted, and the data were synthesized narratively and displayed in Table 2.

The two studies that used a strengthening program focused on plyometric exercises [34] and isokinetic force [33] for the lower limb reported an adherence of 95.83%.

The two studies that opted for a multicomponent exercise program reported similar adherence values ranging from 65% [16] to 67% [31].

Finally, the study that used an exergaming protocol using Nintendo Wii obtained 100% adherence [30].

A chi-square test comparing adherence across the three intervention types revealed no statistically significant differences (χ^2^ = 0.667; *p* = 0.716).

### 3.3. Dropouts

Figure 3 shows the meta-analysis of the dropouts. The estimated average log odds ratio based on the random-effects model was −0.12 (95% CI: −0.69 to 0.45) with a small heterogeneity (I^2^ = 19.66%), which means a relative reduction of 11.2% in the exercise group compared to the control group. However, the meta-analysis was not statistically significant (*p* = 0.501).

Figure 4 summarizes the proportion meta-analysis of the dropouts divided by type of exercise. The combined proportion of dropouts in multicomponent exercise was 18.4% (95% IC 10.6–30.1), while the dropout rate for strength exercise and virtual reality was much minor, 8.6% (95% IC 3.8–18.8) and 8.7% (95% IC 0.7–27.4).

Table 3 displays the dropouts in the control group and the exercise group due to each reason in each study. In this table, the lack of interest (34 dropouts) and the death of the children during the intervention (28 dropouts) were the main factors that influenced dropouts. However, these data were only absolute values.

Figure 5 shows the multivariate meta-regression analysis. This revealed that the factors identified as causes of dropouts didn’t moderate dropout rates. The schedule and the lack of compliance were the most influential factors with the highest risk ratio values, but the results were not statistically significant. 

## 4. Discussion

As far as the authors are aware, this is the first study to investigate the factors influencing adherence and dropout rates in exercise programs for children with acute lymphoblastic leukemia. Nevertheless, some previous authors have conducted similar studies in other pediatric populations, such as Guijo et al. [36], who aimed to determine whether the characteristics of physical activity interventions can influence adherence rates in children with obesity.

Among the studies analyzed, the adherence was higher in the exergaming protocols (100%) [30] followed by the strengthening protocols (95.83%) [33,34], while this adherence suffered a great decrement in multicomponent exercise programs (65–67%) [16,35]. These findings were consistent with existing literature, which suggests that engagement and exercise program design significantly influence adherence rates in pediatric oncological patients [37]. Furthermore, the greater adherence rates found in the exergaming programs of this review agreed with previous studies in which game-based interventions demonstrated a great adherence in children due to the motivation given by the social interaction, the amusement, and the playful component [18]. Likewise, the high rates of adherence in strengthening protocols were also consistent with previous studies, in which children and families find the compliance of strength training feasible and motivating due to the progressive structure, the simplicity, and the tangible outcomes [38]. In contrast, multicomponent protocols achieved a low adherence because they are more structured and complex, the sessions are longer, and there is a need for several adjustments to avoid fatigue due to the mix of several types of interventions, so the motivation decreases [39].

Following the analysis of the adherence, it is relevant to take into consideration that the characteristics of the intervention are not the only factors that influence adherence; the environment in which the program is developed can even be a key factor. In this study, a great adherence was found among the supervised interventions [30,33,34], a fact that is consistent with previous research, in which the supervision, the structure, and the professional support of the physiotherapist were found to be relevant factors that help the maintenance of participation in the active rehabilitation programs [40,41].

Having discussed how adherence varies by exercise modality and supervision, attention is now turned to dropout rates to further explore factors affecting sustained participation.

In this regard, the meta-analysis showed a relative reduction of 11.2% in the dropout rates in the exercise group compared to the control group, but this outcome was not statistically significant. This finding could be explained by the fact that, in most of the analyzed randomized controlled trials, the control group was engaged in physical activity as well but under less structured or supervised conditions. Consequently, children and their families in the control group also increased their awareness of the importance of engaging in physical activity, which could explain the similar dropout rates, as previously stated by other authors [42,43] who found that the provision of any exercise stimulus to control participants raised baseline engagement. This effect may result from the Hawthorne effect, whereby individuals modify their activity for being monitored; increased contact with healthcare professionals, exposure to educational materials, and parental involvement, as previous researchers had highlighted [44].

Nevertheless, when the dropout rate was analyzed according to the type of intervention, the findings were clearer and statistically significant, with small values (8.6% and 8.7%) for strength exercise and virtual reality programs, and high values (18.4%) for multicomponent exercise programs. These findings suggest that the exercise modality has a potential role as a moderator of adherence, influencing participant retention, confirming our previous findings about adherence, and, according to Vancampfort et al. [45], who also indicated that certain exercise modalities are associated with lower dropout rates in the pediatric population.

Having reviewed dropout rates across groups and exercise modalities, the focus now turns to the underlying reasons for these withdrawals.

Further analysis of the reasons for dropouts in active physiotherapy interventions among pediatric oncological patients revealed that the lack of interest was the main factor, causing 34 dropouts among the studies analyzed, which agrees with previous qualitative studies in which children and their families identified the lack of interest or motivation as a major barrier to complete the studies as they may not fully understand the purpose of the intervention or experience negative attitudes toward exercise [46]. This generalized lack of interest among children could be influenced by psychological factors such as anxiety, depression, and low self-efficacy, which, together with physical symptoms as cancer-related fatigue, could decrease intrinsic motivation to maintain participation in exercise-based programs, as Adamovich et al. [47] previously highlighted.

The second most influential factor was death, causing 28 dropouts, with a balanced number of dropouts among the exercise group and the control group. This finding reflects the clinical severity of the population rather than a modifiable barrier to perseverance throughout the intervention. This finding is consistent with previous studies that have highlighted mortality as one of the main causes of dropouts in medium-long-term exercise interventions, noting the inevitability of mortality in this context [48]. In addition to this factor, another non-modifiable determinant, such as concomitant diseases, was identified as an influential variable, emphasizing the outcomes found by other researchers, who noted the presence of comorbidities as a significant impediment to consistency in participation in physiotherapy interventions [49].

Despite the identification of these causes, meta-regression analysis showed that none of them could be considered a moderator because the results were not statistically significant. This fact could be explained by the huge heterogeneity of the studies included, which, together with the individual characteristics of the patients and the limited number of included studies, makes it difficult to determine consistent moderators, as other authors have noted [50].

Given the multiple factors contributing to loss to follow-up among pediatric oncology patients in active physiotherapy interventions, there is a clear need for designing effective, sustainable, and individualized exercise programs for pediatric oncology patients, considering physical symptoms, comorbidities, and psychological barriers to optimize adherence. Practical recommendations for such programs include prioritizing engaging exercise modalities, such as exergaming and strength training, to enhance muscle condition and overall fitness while maximizing adherence to the programs [16,30]. Moreover, ensuring supervision by professional physiotherapists is essential to promote safety, correct technique, and also increase engagement [42]. Additionally, incorporating motivation strategies, including goal setting, positive feedback, social interaction with team dynamics, and parent involvement, to further reinforce the engagement and the sustained participation across the full period of the physiotherapy program [51].

Therefore, future studies should focus on enhancing adherence through engaging and supervised interventions since it may also nurture resilience, empowering children and families to maintain active participation despite treatment-related challenges.

### Strengths and Limitations

The main strength of this study is its novelty since it is the first time that a systematic review with meta-analysis of factors influencing adherence in active physiotherapy programs is carried out in such a specific population as children suffering from acute leukemia. Furthermore, another strength of this study was the inclusion of the meta-regression, which enhances methodological rigor and the robustness of the conclusions.

Moreover, the conscious assessment of the solid validity of the randomized controlled trials included using the PEDro scale, and the risk of bias assessment tool could also be considered as a strength of this study. Likewise, this study provided two detailed meta-analyses, including a traditional one and a proportional meta-analysis, which confer greater value to the outcomes found.

However, several limitations should be mentioned. Firstly, the scientific literature lacks randomized controlled trials that analyze the adherence and dropout rates in exercise-based programs. Secondly, the heterogeneity of the active physiotherapy protocols used, as well as the individual characteristics of the patients, also limits the consistency of the drawn conclusions. Thirdly, the lack of statistical significance in some of the results exposed determines that further research is needed in order to confirm the hypothesis launched. Finally, instead of the novelty being noted as a strength, it could also be considered as a limitation since it limits the generalisability of the conclusions, as the evidence is not strong enough.

## 5. Conclusions

The present study confirms that the heterogeneity of the studies in children suffering acute leukemia meant that no factor could be identified as a moderator of dropouts in active physiotherapy interventions. However, exercise modality stands out as a potential moderator of adherence, with exergaming protocols and strength-based exercise programs demonstrating great adherence. Nonetheless, this study highlighted that future trials should focus on standardized adherence measures, larger samples, and tailored motivation strategies to confirm the results of this review and to allow clinical implementation.

## Figures and Tables

**Figure 1 healthcare-13-02766-f001:**
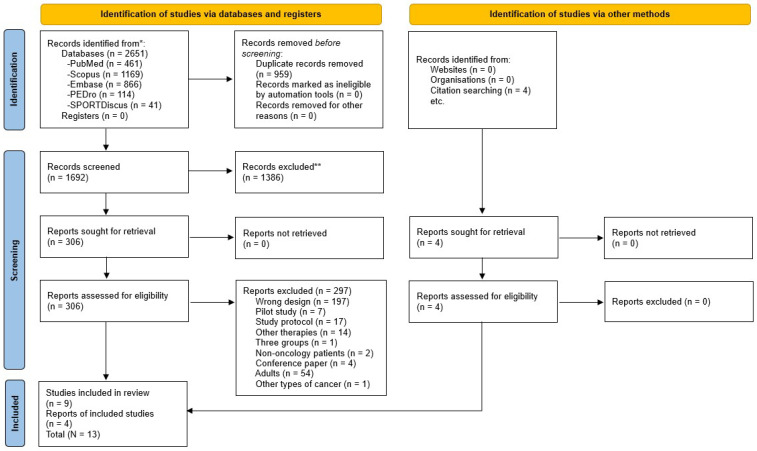
Flow diagram PRISMA 2020 for systematic reviews. *** Records identified in each database. ** Records excluded by Title and Abstract**.

**Figure 3 healthcare-13-02766-f003:**
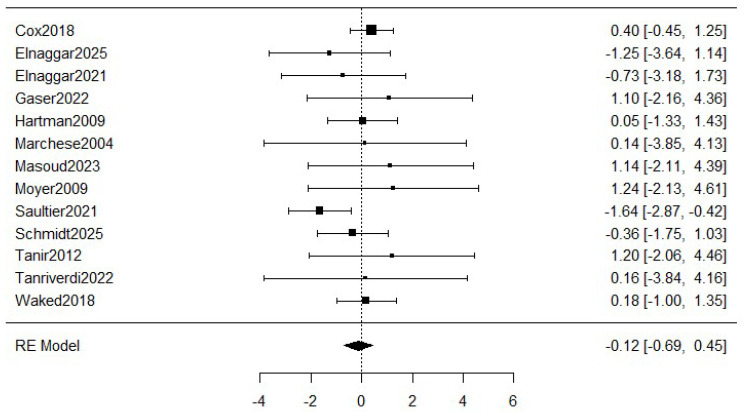
Forest plot for meta-analysis of the dropouts comparing exercise groups and control groups. The figure analyses thirteen studies. Concretely, it represents the mean difference of a study represented by squares and their 95% CIs represented by horizontal lines. The size of each square reflects the study weight. The diamond at the bottom represents the pooled mean difference, calculated using a RE (random effects) model [5,8,16,26,27,28,29,30,31,32,33,34,35].

**Figure 4 healthcare-13-02766-f004:**
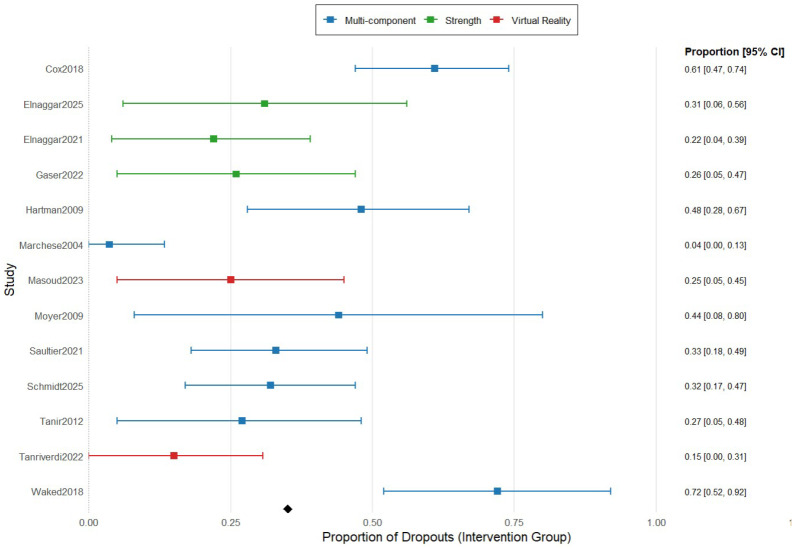
Forest plot for proportion meta-analysis of the dropouts comparing the type of exercise in the intervention group. The figure analyses 8 studies using multicomponent exercise, represented in blue color, 3 studies using strength exercise, represented in green color, and 2 studies using virtual reality-based exercise, represented in red color. The figure shows the point estimates of each study represented by squares with their 95% CIs represented by horizontal lines, and the pooled proportions represented by the diamond. The analysis was conducted using a random-effects model [5,8,16,26,27,28,29,30,31,32,33,34,35].

**Figure 5 healthcare-13-02766-f005:**
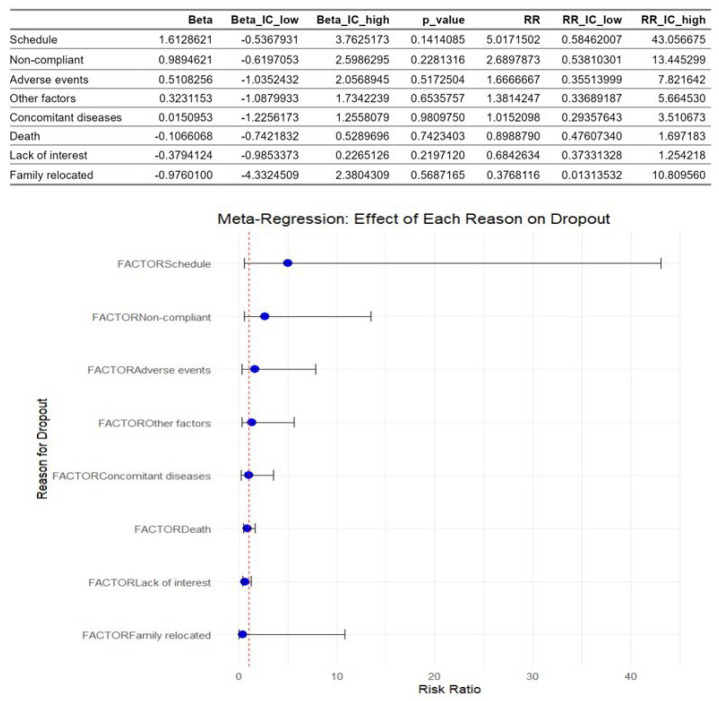
Meta-regression of factors associated with dropouts in children with acute lymphoblastic leukemia during an active physiotherapy program. The figure represents 13 studies. The figure shows the regression coefficients (Beta) for each factor with their 95% CIs and p-values, as well as the corresponding risk ratios (RRs) represented by the circles and their 95% CIs represented by the horizontal lines. The analysis was conducted using a random-effects model, with the natural logarithm of the RR as the dependent variable [5,8,16,26,27,28,29,30,31,32,33,34,35].

**Table 3 healthcare-13-02766-t003:** Identification of factors influencing dropouts.

Reference	Group	Factors Influencing Dropouts								Total Dropouts
		Death	Non-Compliant	Lack of Interest	Concomitant Diseases	Schedule	Family Relocated	Adverse Events	Other Factors	
Cox et al. (2018) [26]	**CG**	1	0	7	3	0	1	0	1	13
	**EG**	2	1	4	4	0	0	0	6	17
Elnaggar (2025) [33]	**CG**	0	1	2	0	0	0	0	2	3
	**EG**	0	1	0	0	1	0	0	0	1
Elnaggar and Mohamed (2021) [34]	**CG**	0	0	0	0	0	0	0	2	2
	**EG**	0	0	0	0	1	0	0	0	1
Gaser et al. (2022) [16]	**CG**	0	0	0	0	0	0	0	0	0
	**EG**	1	0	0	0	0	0	0	0	1
Hartman et al. (2009) [5]	**CG**	2	0	2	1	0	0	0	0	5
	**EG**	2	0	2	1	0	0	0	0	5
Marchese et al. (2004) [29]	**CG**	0	0	0	0	0	0	0	0	0
	**EG**	0	0	0	0	0	0	0	0	0
Masoud et al. (2023) [30]	**CG**	0	0	0	0	0	0	0	0	0
	**EG**	0	0	1	0	0	0	0	0	1
Moyer-Mileur et al. (2009) [8]	**CG**	0	0	0	0	0	0	0	0	0
	**EG**	0	0	1	0	0	0	0	0	1
Saultier et al. (2021) [27]	**CG**	2	0	12	0	0	0	0	0	14
	**EG**	1	0	3	0	0	0	0	0	4
Schmidt-Andersen et al. (2025) [31]	**CG**	2	1	0	0	0	0	3	0	5
	**EG**	2	0	0	0	0	0	2	0	4
Tanir and Kuguoglu (2012) [35]	**CG**	0	0	0	0	0	0	0	0	0
	**EG**	1	0	0	0	0	0	0	0	1
Tanriverdi et al. (2022) [28]	**CG**	0	0	0	0	0	0	0	0	0
	**EG**	0	0	0	0	0	0	0	0	0
Waked and Albenasy (2018) [32]	**CG**	7	0	2	0	0	0	0	0	9
	**EG**	5	5	0	0	0	0	0	0	10
**TOTAL (per group)**	**CG**	**14**	**1**	**23**	**4**	**0**	**1**	**2**	**5**	**50**
	**EG**	**14**	**8**	**11**	**5**	**2**	**0**	**3**	**6**	**49**
**TOTAL**	**Both**	**28**	**9**	**34**	**9**	**2**	**1**	**5**	**11**	**99**

## Data Availability

No new data were created or analyzed in this study. Data sharing is not applicable to this article.

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
