# Peer review of "Identifying Key Factors in Adherence and Dropouts in Active Physiotherapy in Children with Acute Leukemia: A Systematic Review with Meta-Analysis and Meta-Regression"

_healthcare, 2025, doi:10.3390/healthcare13212766_

Round 1
Reviewer 1 Report
Comments and Suggestions for Authors
Abstract:
- ensure consistency in reporting number of included studies (13 in full text - line 141 vs. 13 in the abstract section)
- current: “dropout rates did not differ significantly (OR = -0.12; 95% CI -0.70 – 0.45).” Please replace with simple statement like “Dropout rates were similar between groups.” Keep exact numbers only if necessary
Introduction:
- rephrase epidemiology sentences: several fragmented sentences (incidence, survival, complications) merge into 2–3 smooth sentences: 1.incidence and survival, 2.complications from treatment
- be more focused on problem statement and clearly separate: 1.treatment complications, 2.role of exercise and 3.problem of adherence/dropout.
MM section:
- eligibility criteria: "children under 18 years old” but later includes adolescents in Table 2 Saultier et al 2021
- you mentionned PEDro and RoB2, but not how disagreements were handled. Add detail: “Two reviewers independently assessed risk of bias; discrepancies were resolved by discussion" or something relevant
Results:
- again for remain: abstract says 12 studies; results text says 13 studies
- keep detailed information in Table 2; summarize key points (sample size, age range, intervention types) in text only briefly
- clarify how many studies reported adherence; standardize reporting (for example: “adherence ranged from X% to Y% in N studies”).
-figures 3–5 referenced, but legends short and not fully self-explanatory. Please expand figure legends: explain abbreviations, variables, and what model was used
Discussion:
- please improve logical flow (1.adherence, 2.dropout, 3.reasons, 4.implications, 5.strengths/limits).
- please clearly compare with pediatric oncology literature and avoid drifting into obesity/adult studies unless directly relevant
- please add practical recommendations: prioritize engaging modalities (exergaming, strength), ensure supervision, and design motivation strategie
Conclusion:
- you stated “Future studies are needed to confirm results.” - it is too generic in my opinion, please be specific, maybe: “Future trials should focus on standardized adherence measures, larger samples, and tailored motivation strategies.”
Reviewer 2 Report
Comments and Suggestions for Authors
Thank you very much for allowing me to review this manuscript, which addresses a very important topic: therapeutic exercise performed by physical therapists for pediatric pathologies.
The manuscript is well-written, with the introduction, results, and discussion presented in clear, easy-to-follow sections.
The references are accurate and aligned with the research question.
Below, I would like to share a series of suggestions and concerns:
Methods
- 3 databases seem few, and since there is active exercise therapy, PEDRo and sportidscuss should be reviewed.
- Including a language barrier as an inclusion criterion is a language bias; it should be eliminated.
Regarding the publication of tables and images, many of them should be included in the supplementary material, as they hinder reading.
Reviewer 3 Report
Comments and Suggestions for Authors
Present the main finding in the title.
In the abstract, cite the type of review as in the title.
Present in the introduction the physical therapy modalities that will be addressed in the review, as well as their description and scientific support for use in these patients.
Methods:
Were duplicates removed manually or by software? Indicate the name of the software.
Why not use PICO?
What is the search window (years)?
Results:
Was there a difference between the type of intervention and adherence?
Discussion:
This should be more extensive and explain each finding with support from the literature.
Round 2
Reviewer 2 Report
Comments and Suggestions for Authors
Thank you for making the suggested changes.
The publication has been significantly improved in every way.
It is ready for publication.
Author Response
Thanks for dedicating your time in reviewing our manuscript. We greatly appreciate your feedback and that you think the manuscript is worthy the merit of publication in this prestigious journal